# Expectations and Experiences of Patients Recently Initiated to Centre-Based Dialysis Treatment

**DOI:** 10.3390/healthcare10050897

**Published:** 2022-05-12

**Authors:** Johan Frederik Lillebø Alsing, Eithne Hayes Bauer, Frans Brandt, Jan Dominik Kampmann

**Affiliations:** 1Department of Clinical Research, Faculty of Health Sciences, University of Southern Denmark, 5000 Odense, Denmark; johan.frederik.alsing@gmail.com; 2Internal Medicine Research Unit, University Hospital of Southern Denmark, Sygehus Soenderjylland, Institute for Regional Health Research, University of Southern Denmark, 5000 Odense, Denmark; ehb@rsyd.dk (E.H.B.); fbk@rsyd.dk (F.B.)

**Keywords:** dialysis, centre-based dialysis, qualitative, focus groups

## Abstract

Existing studies display a huge disparity in terms of the number of patients who regret having engaged in dialysis. Modifiable care processes such as providing sufficient information and education prior to decision-making have been shown to have a greater impact on patient satisfaction. Despite the importance of regret as a measure of the quality of the dialysis decision-making process, few studies have examined regret following dialysis initiation. **Aim:** To explore the expectations and experiences of patients who have recently started centre-based dialysis treatment. **Methods:** A qualitative explorative study of centre-based dialysis patients was performed. Data were collected using focus group discussions of 2–4 patients. The study was guided by interpretive description and thematic analysis was used to analyse data. **Results:** Three focus group discussions were performed. Participants (*n* = 8) consisted of six men and two women aged 54 to 80 years of age with a median age of 72. Three themes emerged from the data: 1. transition from being a non-dialysis patient to becoming a dialysis patient, 2. physical condition following initiation of dialysis treatment, and 3. limitations and social disruptions. **Conclusion:** The initiation of dialysis disrupted daily life in terms of fluctuating fatigue, strict schedules, and time lost. There was a loss of independence, and participants did not view dialysis as an active choice. Nurses may have a significant impact on the perception of dialysis. This study highlights the need for further research to develop interventions to support newly initiated centre-based dialysis patients to transition from non-dialysis to dialysis patients.

## 1. Introduction

Dialysis is the most common treatment form for patients with end-stage renal disease (ESRD) [1]. The total number of patients engaging in dialysis is increasing. It is estimated that by 2030, between four to seven million people will be receiving renal replacement therapy worldwide [2]. However, despite its life-prolonging effect, dialysis is time-consuming, physically challenging, and notably, expensive [1]. Furthermore, dialysis has a significant psychological impact on patients, affecting their overall quality of life [1]. The older population of patients with comorbidities, in particular, may not survive for longer periods or experience justifiable quality of life with dialysis [3,4]. Having said that, patient satisfaction with dialysis varies [5]. Most patients experience dialysis as life changing, with feelings of loss and uncertainty [5]. The difference lies in their ability to overcome new circumstances. Some manage to adapt effectively, while others are unable to accommodate the abrupt life-changing conditions that commencing dialysis treatment entails [5]. In a Canadian study of patients with CKD, 61% reported regret with the decision to start dialysis [6]. On the contrary, in a study from the United States, the prevalence of regret was only 7%. This displays a huge disparity in terms of the number of patients who regretted having engaged in dialysis [6,7]. While demographic factors are, to a lesser extent, associated with regret, modifiable care processes such as providing sufficient information and education prior to decision making have been shown to have a greater impact on patient satisfaction [1,7]. Despite the importance of regret as a measure of the quality of the dialysis decision-making process, few studies have examined regret following dialysis initiation [8]. Regret is defined as “a feeling of sadness about something sad or wrong or about a mistake that you have made, and a wish that it could have been different and better” [9]. Therefore, the goal of this qualitative research study was to gain further insight into regret by exploring patients’ expectations and experiences following dialysis initiation among centre-based haemodialysis patients. The term centre-based haemodialysis refers to regularly scheduled in-hospital haemodialysis treatment. This involves two to four trips weekly to the nearest dialysis treatment centre for three to four hour dialysis sessions. This study adds to the body of knowledge on experiences, expectations, and preferences regarding the choice of treatment for dialysis patients. Furthermore, it may guide physicians and nurses to a better understanding of patients’ needs regarding decisions and the initiation of dialysis treatment. Clinical implications may involve implementing adjustments to modifiable factors, which may result in increased patient satisfaction with the choice of treatment.

### Aim

To explore expectations and experiences of patients who recently initiated centre-based dialysis treatment.

## 2. Materials and Methods

A qualitative explorative study based on interpretive description (ID) was performed among patients receiving dialysis treatment. Interpretive description, which is a non-categorical research approach, is an appropriate design and method for this study, as it takes into account the researcher’s background and the clinical setting [10]. Within this framework, focus group discussions guided by Malterud were performed to collect data, and inductive thematic analysis guided by Braun and Clarke was used to analyse data [11,12]. Findings are reported in accordance with the consolidated criteria for reporting qualitative research (COREQ) which is a checklist developed to increase transparency in reporting findings in qualitative research [13]. 

### 2.1. Setting

The dialysis ward is located in the Hospital of Southern Jutland in the Region of Southern Denmark and receives patients from four municipalities in the region. At the time of writing this paper, 68 patients were receiving haemodialysis treatment at the centre. Opening hours are from 7.30 a.m. to 9.30 p.m. on weekdays, 7.30 a.m. to 2.30 p.m. on Saturdays, and 2.30 p.m. to 9.30 p.m. on Sundays. Patients can receive treatment from morning to afternoon or from afternoon to evening. Data were collected in a haemodialysis ward at the centre while patients were receiving treatment. 

### 2.2. Data Collection

Three focus group discussions lasting 50–60 min were conducted to explore common experiences and differing opinions through discussions in a sample of the population of interest. A pilot interview was performed to validate the interview guide, which was found to be appropriate and was entered into the overall dataset. The interview guide (Appendix A) was based on open-ended semi-structured questions. The first author functioned as the moderator while EHB acted as an observer taking field notes. The focus groups were conducted in accordance with patients’ dialysis treatment schedules in order to prevent unnecessary distress and transport. Interviewing during dialysis was also considered to increase the rate of acceptance of potential participants. In order to ensure confidentiality within the focus groups, participants were encouraged to respect the confidentiality of the group. Nurses left the room during data collection and when they were required to attend to a participant, they did so without interrupting the whole group. Furthermore, nurses have a duty to observe patient confidentiality. Citations to support findings were translated from Danish to English by EHB, who is a native English speaker and fluent in Danish.

### 2.3. Population

Patients were invited to participate in the study using purposive sampling regarding age and sex via a secretary from the dialysis centre who functioned as a gatekeeper. Inclusion criteria consisted of patients who had initiated centre-based dialysis treatment less than two years from the start of the study. All potential participants accepted the invitation to take part in the study.

### 2.4. Ethical Considerations 

All participants received oral and written information about the purpose of the study. Informed consent and demographic data were obtained prior to the focus group discussions. Participants were informed of the possibility of withdrawing consent without further consequences. The choice of the dialysis centre as the location for all three focus groups was a pragmatic choice. However, the location is a place of work, and the focus group discussions took place while the participants were receiving dialysis treatment. Some participants required attention during the discussions, and this may affect the data collected. Both the moderator and observer dressed in plain clothing to prevent a mismatch of authority between participants and researchers. The first author is not an employee of the hospital. The second author, who was an observer during data collection, is employed in a different unit in the hospital. There was no interaction between participants prior to the discussions. All focus group discussions were recorded, and data were stored securely. This study was approved by the Danish Data Protection Agency (Journal no.: 21/57866). The study complies with WMA’s Declaration of Helsinki [14]. 

## 3. Results

### 3.1. Participants

Participants (*n* = 8) comprised of six men and two women in the age group 54 to 80 years with a median age of 72. Participant demographics are illustrated in Table 1 below.

### 3.2. Data Analysis

All focus group discussions were transcribed verbatim, and data were anonymised. Inductive thematic analysis was used to analyse the dataset. Data were coded, categorised, and divided into themes and subthemes.

### 3.3. Themes

Three themes were identified: (1) limitations and social disruptions, (2) physical condition following initiation of dialysis treatment, and (3) transition from the last outpatient consultation to initiation of dialysis. Themes and subthemes will be presented supported by citations. A brief overview of themes is presented in Table 2 below.

In line with the method and design of interpretive description, each theme is summarised and results are described supported by citations, and initial tentative interpretations are drawn.

#### 3.3.1. Transition from Non-Dialysis to Becoming a Dialysis Patient

There were many different expectations regarding dialysis and its effect on daily life. In summary, all participants expressed gratitude for the opportunity to initiate dialysis treatment. Many of the participants described an awareness of their prognosis with the eventual need for dialysis and were optimistic about the promise of symptom reduction and regaining functions. Still, many expressed receiving the news that dialysis was inevitable as abrupt and were shocked.


*Yes, I knew that I’d have to have dialysis, but not for another ten years. That was what I was told. I have to say, that was the longest trip from Soenderborg and home in my life. Damn it! I’ll tell you that can be done in another way! At least a warning or something. Something or other, so you’re prepared when you come here.*
(Participant No. 3)

One participant described how the expected time for initiation of dialysis was off by several years and that he was under the impression that he was coming to a regular check-up. Others described similar experiences; where they could not understand how their kidney function had declined so quickly. Some participants showed awareness of the progression of their disease.


*Yeah, in my case I’d known for a long time that my kidneys were getting worse and worse, but … it’s a death sentence. We’re thinking that now the best is over.*
(Participant No. 2)

Pessimistic thoughts regarding the treatment and life situation were not uncommon, as many viewed dialysis as simply postponing death. Although most participants pointed out that others starting dialysis should not be scared, they all admitted that they had fears or worries themselves. Typical thoughts were fear of the unknown, whether they can live with the treatment and for how long. Many worries dissipated as the participants started treatment, although new worries emerged. On the other hand, participants who either started dialysis acutely or with little previous notice reacted differently to the above.


*Yeah, well, my expectations were zero; it came so quickly. I went for some examinations and all of a sudden; “he has to start dialysis!”*
(Participant No. 1)

The abrupt onset of treatment appeared to allow little time for expectations, increasing the initial shock. Emotional displays during the discussions gave the impression of not having come to terms with one’s current life situation. Participants who experienced an abrupt onset of treatment also expressed that they felt the decision was already made for them.

Some participants expressed experiencing frequent thoughts of suicide as a result of initiating dialysis.


*But, it’s not something I consider on a daily basis. Well, that’s not how you should understand it. But, the thought does occur to me. What if - boom – job done! Then my wife can come out and travel again.*
(Participant No. 2)

This appeared to arise from the current life situation with the progression of illness, the potential failure of treatment, and the abruptness of dialysis. In addition, participants expressed an underlying feeling of guilt as they felt like a burden to their families. This was also a common reason for not choosing home dialysis. Social support from relatives was an important part of coping with the progression of the illness. Some participants described acceptance of the loss of functionality, and the possibility for discussing feelings, thoughts, and choices. Excellent service was also mentioned as something that eased the transition. However, there seemed to be a lack of peer support amongst patients receiving dialysis treatment together.


*Yeah, when we’ve been in beside each other, then we’ve been three to four people. But, well, we don’t’ talk to each other, not like we two are doing now.*
(Participant No. 1)

While participants described how swapping thoughts and experiences among each other was good, there appeared to be little interaction with other dialysis patients both during and outside of their treatment schedule. During dialysis, they were mainly confined to their own space; napping, eating, or watching TV on their monitors. When asked whether the participants regretted their choice of initiation of dialysis, all the participants seemed grateful for the treatment. However, they also implied that there was really no choice.


*It’s as he says. Is there a choice? If there isn’t any kidney function and it helps to get the blood rinsed, so you can live on in this way. Yeah, then that’s what you choose!*
(Participant No. 5)

The participants clearly expressed that the choice that they were facing was more a question of whether to live or to die. They also expressed that they were unable to decide over their own time due to the set scheduling of the treatment. Remaining choices seemed to acquire more importance as they increased their sense of autonomy. This was implied, e.g., in their gratitude for the possibility to choose their meals.

#### 3.3.2. Physical Condition following Initiation of Dialysis Treatment

All participants frequently mentioned fatigue and its impact on their ability to maintain a fulfilling lifestyle. The level of energy changed for everyone following the initiation of dialysis treatment. However, participants’ perceptions of whether dialysis was beneficial to them differed.


*I also agree with that. It was difficult before I started. But, it’s come gradually. It didn’t come all at once. It’s come slowly but surely over some months. So, the treatment works.*
(Participant No. 3)

Participants with a high degree of fatigue at the time leading up to dialysis treatment experienced greater ease of the symptom. They described a slow and consistent decrease in fatigue after dialysis initiation, which subsequently allowed them to regain some of their daily life routines. These participants also expressed a greater level of contentment with the dialysis treatment. On the other hand, there were those who had few symptomatic complaints prior to dialysis start. 


*For me, it’s been a huge challenge. Yeah, there are a lot of things that you can’t do anymore because you don’t have the energy. You can’t just say; ”now, I’m going to…” what do I know..” take a trip or something or other”. You have to come in here.*
(Participant No. 5)

These patients described how the initiation of treatment reduced their energy and this gave rise to more extensive disruptions to the activities of daily life. Some also described a slow and consistent decrease in fatigue the longer they remained in dialysis. All patients noted a fluctuation of fatigue in accordance with the rounds of dialysis. 


*Yes, but I think, actually, that I’m tired the day that I’m in dialysis mostly. But, I have three weekdays there. And, then, I come to on Wednesday and then I feel good.*
(Participant No. 3)

This pattern had a strong tendency to peak at the end of the dialysis rounds, especially amongst participants in the earlier rounds of the week; those receiving treatment earlier in the day tended to take naps after treatment and regain energy. Some also indicated that the afternoons after dialysis were wasted due to tiredness. On the contrary, those receiving dialysis in the afternoon expressed the early hours of the day as useful.

Most of the participants experienced alleviation from some of their worries after starting dialysis. On the other hand, new worries emerged. Especially, worries regarding the functioning of the arteriovenous fistula were frequently brought up.


*Just so long as it works! Because sometimes we get cannulated many times, and it’s not particularly nice. So, I hope for the best. But, that’s yet another thing. So, how many things are there?*
(Participant No. 2)

Fear regarding the continuous decrease of blood flow of the fistula and possibly, its eventual failure was a common theme among participants. For some, their fistula was the last available access point for haemodialysis, and they were incompatible with peritoneal dialysis. Annoyance with painfully repeated cannulations was also expressed. Contentment with healthcare personnel and especially their ability to cannulate had a positive impact on participants’ experiences of their treatment sessions.


*Of course, in the beginning, there were several in to jab me. Then it took some time and you built a relationship. So, that period was sad. But, after a while, it all came, all of them. And, they’re all very good at cannulating. So, you actually get help no matter who’s here. That’s definitely a good help for me.*
(Participant No. 4)

Several participants described affection for the nurses, which could be credited towards their professional competencies. This, together with the healthcare personnel’s open-minded, accommodating, and friendly attitude, provided a strong sense of trust and security, as patients tended to describe feeling at home on the ward.

#### 3.3.3. Limitations and Social Disruptions 

Both disruptiveness of the enforced dialysis schedule and time spent on the treatment, in particular, were mentioned often and predominantly among those with pre-existing active lifestyles.


*I know that the nurses were kind to tell us; “No, no, no! There are lots of opportunities but if you’ve lived an active life then it’s difficult.*
(Participant No. 2)

Frustration appeared to be caused by the intrusiveness of the treatment on daily life. Especially, the social aspect of life became limited by the disruption caused by dialysis.


*Can’t go to parties and out dancing. No, forget it! I don’t do that either. That’s over!*
(Participant No. 1)

The many hobbies and social activities, and especially, those requiring active participation were no longer an option. Participants described the cause as multifactorial but the most common reasons were lack of energy and time, and a sense of shame over their dialysis in relation to their social circle. Most participants were already retired before dialysis initiation. 


*We had already retired early, so there were problems before we started here.*
(Participant No. 3)

Patients described that they had reached pension age, taken ill-health early retirement, or had pre-existing illnesses. A few others were working before dialysis but found the new routines incompatible with their existing line of work. Some described how they adjusted by gaining flexible jobs which they still described as too challenging. 

Traveling was a common topic among all participants, which they described as severely limited.


*We were used to taking a weekend away somewhere in Denmark once a month and driving all over the north of Germany. We can’t do that either when we’re in dialysis. (…)That’s why we leave on Saturday. Saturday morning and home Sunday.*
(Participant No. 8)

The rigid schedule for dialysis was described as restrictive on the ability to visit family, friends, and go on vacation. Not only were the participants affected by this but also their spouses and related family. All of the patients were familiar with guest dialysis, but almost none actually described it as a possibility for them.


*Well, I wouldn’t dare to go anywhere else. Well, for longer than the three days I have per week. That’s also because I’ve had a lot of trouble with, e.g., the fistula I have. I feel safe here. They know what they’re doing. *
(Participant No. 5)

This was partially due to fear of the unknown. They described a lack of trust for other dialysis centres and were uncomfortable with unfamiliar routines. Some expressed concern about poorer levels of hygiene and professionalism in foreign centres describing the fear of infections or damage to the fistula. 

Transportation was also important for the participants. They expressed being frequently annoyed about how much time they wasted on traveling.


*It’s not just here that time goes, it’s that you have to get ready and drive from home. And, then, home again, and before you notice, it’s what? Half-past three?*
(Participant No. 8)

Annoyance was expressed from time spent preparing for transport and the transportation itself. Taxis were sometimes delayed, with one participant experiencing up to an hour’s delay. Despite describing transportation as a waste of time, participants expressed gratitude for traveling free of charge. 

## 4. Discussion

### 4.1. Discussion of Participants

The participants had approximately 182 days of dialysis at the time of the discussions. The lowest was with 24 days. Only one of the participants had more than 1 year of experience with dialysis. This may affect the answers in regard to the intrusiveness of the treatment, as some may yet have to come to terms with the abrupt life changes. However, patients who were recently initiated to dialysis treatment may recall the transition from non-dialysis treatment and its effects more clearly. 

### 4.2. Discussion of Results

This study aimed to explore and understand expectations and experiences from the patient’s point of view. The main findings are mentioned below and are discussed under the following themes: transition from non-dialysis to becoming a dialysis patient, physical condition following initiation of dialysis treatment, and limitations and social disruptions. 

Despite knowledge of inevitable kidney failure, dialysis patients were unprepared when receiving the news of imminent dialysis initiation;Patients experienced feelings of guilt and being a burden on their families;The decision to initiate dialysis treatment was regarded by patients as more of a life-or-death decision, rather than a choice of treatment;Some patients felt that they have little or no say in decisions regarding treatment;Dialysis patients in this study did not regret initiating dialysis;Patients experienced fatigue following initiation of dialysis differently;Nurses were an essential part of dialysis patients’ lives, especially in their ability to cannulate;Patients were regularly concerned about the integrity of their vascular access to dialysis, e.g., the fistulas declining function, possible failure, and consequences;There was a perceived lack of peer support among dialysis patients receiving dialysis together;Most patients experienced dialysis as incompatible with an active lifestyle;Dialysis treatment prevented patients from participating in social activities;Travelling for leisure was severely restricted due to strict dialysis schedules;Worries related to guest dialysis prevented patients from taking vacations and availing themselves of dialysis on vacation; andPatients experienced frustration in relation to time lost, not only on treatment but also on preparation for traveling and delays.

#### 4.2.1. Transition from Non-Dialysis to Becoming a Dialysis Patient

##### The Lack of Choice and Independency

Participants displayed a lack of autonomy regarding both the disruptiveness of the treatment and the lack of a subjective feeling of not being responsible for decisions regarding treatment. The choice regarding treatment initiation was more an existential question of whether to live or die, rather than an active choice, as mirrored in findings by Russ et al. [15]. Many implied that healthcare workers made the choice for them. This seemed to affect their ability to accept their current life situation. Similarly, a Swedish study found that patients’ sense of independence and normality was lost with dialysis [16]. Likewise, another study showed that those who deemed the nephrologist’s opinion to be vital in the choice of treatment had an increased experience of regret compared to those who perceived the decision to be made by themselves [1]. Therefore, a more proactive inclusion of patients and families in the decision-making process prior to initiation of dialysis may increase contentment with treatment.

##### Guilt and Its Impact

Many of the participants experienced guilt about being a burden to their families and suicidal thoughts were connected to releasing one’s spouse from this burden. Being a burden was perceived as a major factor for not choosing home dialysis, so as not to place undue responsibility on families. However, a recently published study differs, concluding that lack of a care partner was one of the main driving forces for opting out of home dialysis [17]. Again, proactive participation of patients and families at an early stage in decision-making may help to reduce feelings of guilt and being a burden. 

#### 4.2.2. Physical Condition following Initiation of Dialysis Treatment

##### Fatigue

Participants experienced a fluctuation of fatigue peaking after dialysis sessions. This had a great impact on restoring the part of the day after treatment, and some considered the whole day as occupied with dialysis. Similar findings in regard to the fatigue pattern and its impact were reported in another study [5]. The participants with afternoon treatment were, however, less impacted by fluctuations, as they could sleep shortly after coming home. It can, therefore, be questioned whether afternoon dialysis would ease the impact of post-dialytic fatigue compared with morning dialysis.

##### The Importance of the Nurses, Peers, and Concerns about the Fistula

Nurses play an essential role in dialysis patients’ lives. They are considered friendly and service-minded, providing a substantial amount of comfort and security. What participants appreciated most was nurses’ abilities to cannulate fistulas, thus avoiding pain and the need for repeated attempts. Dissatisfaction was expressed when nurses went on vacation and were replaced by new or less skilled nurses. Patients were regularly concerned regarding the vascular access to dialysis and its declining function, possible failure, and consequences thereof. Results implied that patients considered that good cannulating technique and maintaining high hygiene standards can prevent damage to the fistula and infections. However, it is important to point out that as a single-centre study, this finding may be based on the attitude of nurses in this location alone. With that said, a similar study concluded on the importance of the nurses, and how their skills and interactions affect patients’ perceptions of dialysis [18]. It is, therefore, noteworthy that a trusting relationship between patients and nurses is built on both an empathetic approach and a high level of skill in cannulation.

The participants admitted little interaction with each other prior to the FDG and the field notes described the participants as more confined to their own space during dialysis. However, on several occasions, participants expressed ease of worry when sharing experiences with other dialysis patients. Therefore, improved peer support may help in the adaptation period by creating a sense of community, sharing both different and similar experiences in the process, and allowing personal reflections. Minimizing fear of the unknown may help to reduce subsequent concerns. However, no references in the literature were found to support this claim.

##### Limitations and Social Disruptions

There was a common experience of dialysis being disruptive with regard to social aspects, traveling, and other activities. This was due to the loss of time and the strict dialysis schedule. Transportation and preparation were also factors in the experience of time lost or wasted. Together with fatigue, most participants deemed days with dialysis as wasted which is similar to findings in several other studies [19,20]. The impact of this disruption varied, as those with a previously active lifestyle seemed less satisfied with their current life situation, finding dialysis incompatible with an active lifestyle. On the other hand, those with inactive lifestyles were more content and less affected by the change. Although all patients were aware of the possibility of guest dialysis, none had availed of it, with the main reason being unfamiliarity with other dialysis centres. There was a prevailing notion that they may be subjected to less skilled care and insufficient hygiene standards. Risks of infection and damage to the fistula was of grave concern among participants. However, we were unable to locate evidence to support this finding. Therefore, further research can explore whether this finding is widespread and whether there is a need for counteractive measures.

### 4.3. Strengths and Limitations

The choice of focus group discussions was effective for collecting differing views and experiences. The focus group participants had a level of heterogeneity that allowed expression of both contrasting and similar experiences and thoughts generating discussions. Participants were at ease and did not seem hesitant in sharing vulnerable information. There was rigorous use of COREQ throughout the study.

Limiting factors of the study include some disturbances experienced in collecting data in an active workplace, the physical distance between participants during the focus group discussions, and a non-neutral setting. These factors may have inhibited discussion. Furthermore, in two of the focus groups, only two participants took part, which was partially due to the setting. This could be a limiting factor in relation to holding a group discussion. However, observations from both of these focus groups indicate that both participants from both groups participated in lively discussions contributing valuably to the dataset. The discussion between patients was an important methodological factor in this study, which could not have been achieved through other methods such as individual semi-structured interviews. Data collection was performed in a single centre only, decreasing the generalisability and transferability of the findings. Furthermore, the low number of participants and lack of heterogeneity in relation to sex in two of the focus groups may reflect a limitation of the study. However, as illustrated in the discussion, several findings in this study are mirrored by findings in the literature. For better distribution among sexes, future focus groups studying this topic should attempt to include more women. 

## 5. Conclusions

The purpose of this study was to explore the expectations and experiences of the patients following initiation of dialysis as a measure of regret and highlight adjustable factors that may reduce discontent. Some patients were surprised when told of the need for initiation dialysis as a result of misconceptions with regard to deterioration of kidney function due to the progression of their illness. Going from non-dialysis to becoming a dialysis patient caused a loss of autonomy, changes and fluctuations in fatigue, and a sense of loss of time due to treatment and transportation. Some were accepting of the new reality, whereas others were less content with their new life. Although guest dialysis is a possibility, patients were unwilling to avail of guest dialysis due to fear of being faced with insufficient hygiene, poor service, and less skilled healthcare workers. Nurses appear to have a significant impact on the perception of dialysis. Patients considered the ability to cannulate the fistula without complications a valuable trait amongst nurses in this study. While patients did not express regret in this study, it is clear that the development of interventions to support newly initiated centre-based dialysis patients to transition from non-dialysis to dialysis patients is a necessity and should be the focus of further research, including research into regret between regretting or not starting dialysis versus starting centre-based dialysis instead of home dialysis. Furthermore, the accumulated data may prove useful for hypothesis generation for larger multi-centre qualitative studies or questionnaires.

## Figures and Tables

**Table 1 healthcare-10-00897-t001:** Demographics of patients participating in a study on patient preferences in relation to initiation of dialysis treatment.

Participants N = 8	Sex F/M	Age	Living Alone Yes/No	Distance in km from Dialysis Centre	Total of Days since Dialysis Treatment Initiation
*1*	M	71	No	60 km	240 days
*2*	M	73	No	42 km	60 days
*3*	F	65	No	25 km	575 days
*4*	F	80	Yes	62 km	23 days
*5*	M	54	Yes	25 km	210 days
*6*	M	55	No	51 km	210 days
*7*	M	73	No	36 km	60 days
*8*	M	80	No	16 km	270 days

**Table 2 healthcare-10-00897-t002:** Themes, subthemes, and main findings to be discussed in a study on patient preferences in relation to initiation of dialysis treatment.

Themes	Subthemes	Main Findings to Be Discussed
Transition from non-dialysis to becoming a dialysis patient	- The different expectations - Living with the illness	- Unprepared despite knowledge of eventual dialysis requirement - The optimistic expectations - The pessimistic expectations; “Life as you know it is over” - Those without expectations - Thoughts of suicide - Guilt: reasons for not choosing home dialysis, being a burden - The supportive structure
	- The lack of autonomy	- A decision made for them - No choice, only life or death
Physical condition following initiation of dialysis treatment	- The polarised change in fatigue - The pattern of fatigue	The subjective contrast in experiences of fatigue from initial dialysis initiation to experiences of more energy following months of dialysis: - Improvement - Lack of improvement - A cyclic pattern
	- The fistula - The nurses’ impact	- Worries regarding the fistula integrity - participants’ appreciation of nurses’ technical skills and ability to cannulate the fistula
Limitations and social disruptions	- Social disruption - Dialysis and work - Dialysis and traveling - Why not guest dialysis?	- How the participants were not able to retain previous working and social rituals - How the strict schedule affected the ability to travel, and impacted family - Why the patients did not want to visit other dialysis wards for guest dialysis
	- Transportation	- The experience of transportation to and from the hospital and its impact.

## Data Availability

Data can be obtained from the corresponding author.

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
