# Peer review of "Expectations and Experiences of Patients Recently Initiated to Centre-Based Dialysis Treatment"

_healthcare, 2022, doi:10.3390/healthcare10050897_

Round 1

Reviewer 1 Report

Overall, the research conducted provides important information to clinical practice. Some issues, especially methodological ones, must be addressed. If these issues are addressed, the manuscript will be suitable for publication.

  1. Usually the minimum number of participants in each focus group should be four. Please justify using a lower number of participants, namely given the context, and address this in the limitations.
  2. Regarding this, if it was not possible to guarantee an adequate minimum number of participants in each group, why not performing individual interviews instead (moreover, these may even be more adequate to achieve the aims of this research...).
  3. Delete "The" at the beginning of line 68.
  4. Throughout the text the authors should differentiate between regretting or not starting dialysis vs. starting dialysis instead of home dialysis, since this is only explicitly referred in the discussion (lines 350 ss.).
  5. It is clear if the "Main points to be discussed" in table 2 were determined "a priori" or are the result of the focus groups. Please make it clear, and if necessary replace with a more adequate expression. If they are a result, I suggest including there the issue of home dialysis. 
  6. Please provide Appendix 1 (line 89) for appreciation.
  7. If interviewing was made during dialysis, how was confidentiality assured (lines 91-94).
  8. It is reported that "The first author is not an employee of the hospital". But what about the one taking notes?
  9. Please delete lines 115-121...
  10. Please indicate the distribution of the 8 participants for the focus groups (table 1 and/or respective text).
  11. The fact that all potential participants accepted to take part in the study (line 403) should be referred in the methods.

Reviewer 2 Report

I appreciate the opportunity to review this paper. It presents an analysis of a very important aspect of the care of patients with chronic (CKD) and end-stage kidney disease (ESKD): the transition between late stage CKD and ESKD. The small study size limits the ability to draw broad conclusions but it presents a technique for understanding the patient experience around this transition. The presentation of the paper should be revised because the authors' impressions of the data are incorporated into the results section of the paper and should be separated.

In this section,

3.3.1. Transition from non-dialysis to becoming a dialysis patient 140 

There were many different expectations regarding dialysis and its effect on daily life. 141 In summary, all participants expressed gratitude for the opportunity to initiate dialysis 142 treatment. Many of the participants described an awareness of their prognosis with the 143 eventual need for dialysis and were optimistic about the promise of symptom reduction 144 and regaining functions. Still, many expressed receiving the news that dialysis was inev-145 itable as abrupt and were shocked. 146 

Yes, I knew that I’d have to have dialysis, but not for another ten years. That was what I was 147 told. I have to say, that was the longest trip from Soenderborg and home in my life. Damn it! I’ll 148 tell you, that can be done in another way! At least a warning or something. Something or other, so 149 you’re prepared when you come here. (Participant No. 3) 

The authors provide a summary of their impressions of the participants' feelings about initiation of dialysis. (There were many different expectations regarding dialysis and its effect on daily life. 141 In summary, all participants expressed gratitude for the opportunity to initiate dialysis 142 treatment. Many of the participants described an awareness of their prognosis with the 143 eventual need for dialysis and were optimistic about the promise of symptom reduction 144 and regaining functions. Still, many expressed receiving the news that dialysis was inev-145 itable as abrupt and were shocked. 146 ) It would be preferable to present the quotations from the participants then to summarize the authors' impressions in the discussion. (Yes, I knew that I’d have to have dialysis, but not for another ten years. That was what I was 147 told. I have to say, that was the longest trip from Soenderborg and home in my life. Damn it! I’ll 148 tell you, that can be done in another way! At least a warning or something. Something or other, so 149 you’re prepared when you come here. (Participant No. 3) ) 

A similar pattern is repeated throughout the presentation of the results.

Reviewer 3 Report

The authors aim to understand the expectations and experiences of patients who opt for dialysis and have designed a set of themes to understand patients' perspectives after they enroll for dialysis categorically. Overall, the manuscript is well written. However, there are certain aspects that the authors can work on to improve this manuscript, as discussed below.

  1. There is no information about the type of kidney disease the patients have. Did patients have other co-morbidities? All this information should be added in table 1.
  2. The number of patients enrolled should be increased significantly, especially females.
  3. A statement that nurses have a significant impact on the perception of dialysis (line 425 and other places) should be removed/modified as this study is done only at one center. The data cannot be generalized as at different clinics, the staff is trained differently and may be more vigilant on how to discuss such things with patients. Alternatively, the authors should enroll patients from different clinics to increase the sample size, as suggested above.

Round 2

Reviewer 2 Report

The paper is greatly improved in this revision. This is an important topic and this paper presents a pilot study which could be expanded in Denmark or other countries. 

Reviewer 3 Report

The authors have made necessary changes in the text to discuss the limitations instead of adding data. These changes aptly describe their work with respective shortcomings. Henceforth, no further revision is required.